# Choroidal Melanoma: A Mini Review

**DOI:** 10.3390/medicines10010011

**Published:** 2023-01-05

**Authors:** Noha Soliman, Diaa Mamdouh, Aisha Elkordi

**Affiliations:** 1UCL Institute of Ophthalmology, London EC1V 9EL, UK; 2National Institute for Health Research Biomedical Research Centre for Ophthalmology, Moorfields Eye Hospital NHS Foundation Trust, UCL Institute of Ophthalmology, London EC1V 2PD, UK; 3Royal College of Surgeons in Ireland, Medical University of Bahrain, Muharraq P.O. Box 15503, Bahrain; 4Faculty of Medicine, Kuwait University, Kuwait City P.O. Box 24923, Kuwait; 5Sheikh Jaber Al Ahmad Al Jaber Al Sabah Hospital, Kuwait City, Kuwait; 6King Hamed University Hospital, Muharraq P.O Box 24343, Bahrain

**Keywords:** choroidal melanoma, eye cancer, management, prognosis

## Abstract

Choroidal melanoma is a rare malignant tumour, yet it is the most common primary intra-ocular neoplasm and second on the list of top ten most malignant melanoma sites in the body. Clinical presentation can be non-specific and includes photopsia, floaters, progressive visual field loss, and blurry vision. The tumour is quite often diagnosed clinically during fundus examination; however, the most valued diagnostic tests are A- and B-scan ultrasonography (US). Several factors affect prognosis, including the patient’s age, tumour size, histological features, and presence of metastases. Still, with primary treatment and tight surveillance, around 50% of choroidal melanoma patients metastasise.

## 1. Introduction

The choroid is the layer between the sclera and the retina [1]. The choroid is part of the uveal tract of the eye, which consists of the iris, ciliary body, and choroid [2]. The tumour originates from the melanocytes situated in the uveal tract [2]. Uveal melanoma (UM) is the most common cancer of the eye and is the most common primary intraocular neoplasm in adults [3]. It represents 3 to 5% of all melanomas [4]. Out of these, choroidal is the largest population (90%), while 5 to 8% originate in the ciliary body and 3 to 5% originate in the iris [4].

## 2. Epidemiology

UM incidence differs by sex, race, and country [4]. The incidence of choroidal melanoma is 6 to 7.5 cases per million annually in populations of European descent [5]. The highest incidence is among people with blue eyes and lighter skin [6]. UM is slightly more common in males [7]. Melanomas can arise at any age; however, the majority of cases occur at about 55 years [5]. It has been also stated that the peak for diagnosing UM ranges from 70 to 79 years, with 63 years as an approximate median age of diagnosis [4]. Approximately 50% of patients die from metastatic disease [4].

## 3. Pathophysiology

The mechanism of how UM develops are still unclear; however, it may relate to oxidative damage to pigmented tissues, which are controlled by the type and degree of iris pigmentation. Another theory that has been proposed for the development of UM is the essential genetic sequencing of the pairing bases from adenine-to-cytosine alterations, while ciliochoroidal melanoma has been linked to adenine-to-thymine alterations. UM can also originate from new lesions already present as a pre-nevus lesion [4].

## 4. Genetics

UM is regarded as a sporadic event, though it is related to ocular melanocytosis and dysplastic nevus syndrome [7]. Various genetic mutations have been related to UM [8]. This involves abnormalities in chromosomes 1, 3, 6, and 8 [3]. Monosomy of chromosome 3 is associated with aggressive metastatic disease (5). It is the most common karyotypic abnormality present, accounting for 50 to 60% of cases [3].

Monosomy of chromosome 3 predicates a mortality rate higher than 50%, while disomy of chromosome 3 has a 100% survival rate [9]. Specific alterations in 3p and 1p losses and 8q gain loci are related to high-risk melanomas [5]. Damato and colleagues [10] reported that the 10-year mortality rate for UM is 71% in malignancies with loss of chromosome 3 and 8q gain, 55% in malignancies with monosomy 3, and 0% in malignancies with disomy 3. Another study by Cassoux and colleagues [11] reported that the two-year metastasis-free interval (MEI) is 100% in patients with disomy 3 and normal chromosome 8 but 85%, 82.1%, and 37.1% in disomy 3 and 8q gain, monosomy 3 and normal chromosome 8, as well as monosomy 3 and 8 or 8q gain, respectively.

Genetic studies have emphasised that almost all uveal melanomas contain oncogenic alterations in GNAQ and GNA11 [7]. The development of choroidal melanomas is also associated with breast cancer type 2 susceptibility protein (BRCA2) and BRCA-associated protein 1 (BAP1) genes [5]. Harbour and colleagues (12) reported that alterations in BAP1, which is encoded by a gene on chromosome 3p21 and is essential for neoplasm inhibition, were related to higher distant metastases risk. Tumours with BAP1 mutations are detected in up to half of all UM and often lead to metastasis within 5 years. These tumours are, therefore, considered to have a high risk of metastasis [12]. Splicing factor 3b subunit 1 (SF3B1) mutation often metastasises; however, it can take up to 15 years. These tumours are, therefore, considered to have an intermediate risk of metastasis [13]. UM with an alteration in the Eukaryotic Translation Initiation Factor 1A X-Linked (EIF1AX) gene rarely metastasises. These tumours, therefore, have a low risk of metastasising [14].

From a clinical point of view, RNA-based genetic analysis for choroidal melanoma prognosis, which tests the expression profile for 15 genes seeking to produce prognostic subgroups for metastatic risk, may be performed prior to commencing treatment [15].

## 5. Risk Factors

There is no known cause resulting in UM. It mostly affects individuals with light skin [7]. Other risk factors include iris nevi, freckles, light eye colour, outdoor activities, tanning, and abnormal cutaneous nevi [16]. According to the current literature, choroidal melanoma risk in patients with uveal tract nevus is low [17]. Other conditions that increases the incidence of choroidal melanomas are Nevus of Ota, dysplastic nevi [18], and oculodermal melanocytosis [19]. Ultraviolet exposure is considered to be a risk factor for choroidal melanomas [20]. However, more recent studies contradict this owing to the very low mutation burden observed in the tumour cells [21]. Additionally, a mutational inactivation of the BAP 1 tumour-suppressor gene increases the risk of metastasis in uveal melanoma [22].

## 6. History

The spectrum of symptoms for patients with choroidal melanoma is broad or non-specific and may depend on the tumour location [5]. The most common symptoms are blurry vision (37.8%), flashing lights (8.6%), visual field defect (6.1%), pain (2.4%), and metamorphopsia (2.2%), and patients can be asymptomatic during diagnosing time (30.2%) [23]. Choroidal melanomas can lead to exudative retinal detachment based on the tumour shape [24].

## 7. Physical Examination and Evaluation

It can be challenging to distinguish choroidal melanoma from benign pigmented nevus (Figure 1); moreover, management may be complex as few nevi convert into choroidal melanomas (1 in 8000) (4). Fundus examination, fundus photography, and ocular ultrasound are the main evaluation aspects for ocular tumours [25].

Uveal nevi are generally flat, slate-gray lesions without sharply demarcated margins; their size is limited to about 6 mm in diameter [25,26]. However, there is considerable overlap in size distributions of nevi and indeterminate lesions compared with small melanomas [26,27]. Choroidal melanomas should be suspected if there is orange pigmentation upon fundus examination, tumour thickness > 2 mm, tumour largest basal diameter > 5 mm and hollow acoustic density on the B-scan, subretinal fluid on the optical coherence tomography (OCT), and symptoms of visual acuity loss to 20/50 or worse [28,29].

Upon dilated fundus examination, choroidal melanoma resembles a mushroom collar-button, or dome-shaped tumour with orange lipofuscin pigmentation as well as surface vasculature. Moreover, in melanomas > 4 mm in thickness, it can lead to exudative retinal detachment. Choroidal melanomas are notably pigmented; however, they may vary from darkly pigmented (Figure 2) to amelanotic (Figure 3) [5].

The most valued diagnostic tests are A- and B-scan US. Choroidal melanomas on A-scan US reveal medium to low internal reflectivity with vascular pulsations (Figure 4). Upon B-scan US, choroidal melanoma shows as a mushroom or dome-shaped tumour, indicating extension through Bruch’s membrane. The mass is acoustically silent, indicating a dark appearance inside the tumour (Figure 5). The combination of A and B-scan ultrasonography for tumours > 3 mm in thickness has a 95% accuracy in choroidal melanomas diagnosis [1,5].

Fluorescein angiography (FA) and indocyanine green angiography (ICGA) may be carried out to help distinguish melanomas from underlying pathologies [30]. FA can be performed to assess for intrinsic tumour circulation (double circulation) and extensive dye leakage [5]. ICGA can be used to image the micro-circulation of choroidal melanomas [1]. Melanomas also appear as clumps of hyperautofluorescence on autofluorescence [1].

Furthermore, optical coherence tomography (OCT) and optical coherence tomography angiography(OCTA) have been used recently to capture superficial and deep retinal and choroidal structures non-invasively, providing more accurate details of the lesion [31]. Melanomas on OCT reveal serous retinal detachment, normal retinal thickness, debris on back of retina, and intact photoreceptors [1].

Imaging methods used in the staging of uveal melanomas at baseline and follow-up include liver ultrasound (US); computed tomography (CT) of the head, chest, abdomen, and pelvis; whole body positron emission tomography (PET); or magnetic resonance imaging (MRI) [32]. The function of FNAB is undefined as choroidal melanoma is currently diagnosed clinically with no invasive approaches needed, which may lead to subsequent seeding [1]. However, currently, FNAB is taken more regularly at diagnosis, which aids in developments in the use of gene-expression profiling and cytogenetic analysis [4].

## 8. Differential Diagnoses

The Collaborative Ocular Melanoma Study (COMS) reported a 0.48% clinical misdiagnosis rate, indicating that most intraocular malignancies may only be diagnosed clinically [4]. Shields and colleagues [33] reported that among 400 referrals for evaluation for posterior UM, 8% were diagnosed with haemangiomas (Figure 6), 9.5% were diagnosed with retinal pigment-epithelium hypertrophy (Figure 7), 23.5% were diagnosed with disciform degeneration (Figure 8), and 26.5% were diagnosed with choroidal nevi (Figure 1). It can be also misdiagnosed as melanocytoma (Figure 9), choroidal osteoma (Figure 10), or choroidal metastasis (Figure 11) [34].

## 9. Classifications

Size according to COMS:
-Small: 4.0–8.0 mm diameter and/or 1.0–2.4 mm height;-Medium: 6–16 mm diameter and/or 2.5–10.0 mm height;-Large: >16 mm diameter and/or >10.0 mm height [35].
Based on the cell type: epithelioid, spindle, mixed [34].The tumour, Node, Metastasis (TNM) staging system has several limitations for the classification and prognostication of uveal melanomas, such as the lack of lymph node dissemination, unlike other tumours including conjunctival melanoma [36]. In fact, Cai and colleagues [36] reported that gene-expression profiling is superior in choroidal melanoma classification compared with the TNM classification system.

## 10. Management

UM was historically treated by enucleation [7]. However, treatment of localised UM can currently be classified into eye salvage treatment or enucleation [4]. Eye salvage treatments are generally classified into radiation therapy such as plaque radiation therapy or particle beam radiotherapy, surgical approaches such as local surgical resection, and laser therapy such as transpupillary thermotherapy (TTT) or laser photocoagulation [37].

## 11. Radiation Therapy

Brachytherapy secures a radioactive plaque to the episclera, allowing for transmission of local radiation to the malignancy at a fixed dose [4]. Iodine-125 (I-125) and Ruthenium-106 (Ru-106) are the most common radioisotopes used. I-125 is a γ-emitting radioisotope that penetrates the tissues more deeply than β-radiation Ru-106 [4]. The recurrence rates for 125 I and 106 Ru are 7–10% and 14.7%, respectively [4]. It is associated with good local control; however, it is related to complications such as macular oedema (24.5%), neovascular glaucoma (28.3%), cataracts (44%), and radiation-induced retinopathy (45–67%). Additionally, 58% of UM patients who experience these complications may suffer from moderate loss of vision and poor VA (BCVA < 5/200) [38,39].

Intravitreal anti-vascular endothelial growth factor (IVEGF) following brachytherapy was reported to delay or reduce the loss of vision and macular oedema rates [2]. Brachytherapy is not recommended in UM patients with large basal diameters, extraocular metastases, no light perception vision, and blind painful eyes [40].

Proton beam therapy studies have also reported positive outcomes. A study of all stages of UM revealed that, with a medium follow-up of 5 years, proton-beam-targeted therapy has a 96.4% control rate locally and preserves the eyes in 95% of cases [41].

## 12. Enucleation

The most common surgery for patients with UM is enucleation. It is indicated in the cases of extraocular spread, large tumour diameter, circumferential tumour invasion, and patients suffering from vision loss [39]. Transscleral resection and transretinal endoresection are alternative surgical approaches [4]. Transscleral resection can be used in patients with wide malignancies that are not suitable for radiation therapy and aim for eye salvage treatment [4]. This option has proven to preserve vision; however, it is associated with various complications such as failed vitreoretinal surgery (44–70%), ocular hypertension (21%), retinal detachment (21%), and submacular haemorrhage (16%) [38,39,42]. Transscleral resection has higher recurrence rates in contrast to brachytherapy [43]. Puusaari et al. [44] found that the actuarial recurrence rate was 41% in the resection group vs. 7% in the brachytherapy group five years following intervention. Similarly, Augsburger et al. [45] reported that actuarial 5-year survival rates of 85.2% in the resection group and 81.8% in the brachytherapy group with a rate of severe early vision loss were higher in the resection group. Similar visual outcomes were reported by Kivela et al. and Bechrakis et al. [39]. Bechrakis et al. [39] also reported that the risk of developing neovascular glaucoma was significantly higher in the brachytherapy group in comparison with transscleral resection (33.3% vs. 5.6%). However, there were no differences between the groups in terms of eye retention and mortality rates [39]. In other studies by Puusaari et al. [44] and Caminal et al. [38], the 5-year actuarial enucleation rates were 28% and 29.1%, respectively. Foulds et al. [46] reported that the 5-year overall survival rates in patients with choroidal melanoma who underwent local resection were 79% vs. 54% in patients who underwent enucleation.

## 13. Laser Therapy

Laser therapy, such as transpupillary thermotherapy (TTT) or laser photocoagulation, injects light sensitive mixtures and free radicals to destruct the vascular supplies of the tumour and decreases the recurrences locally [4]. The efficacy of TTT has been reported when used for treating residual choroidal melanomas as well as adjuvant treatment following brachytherapy [47,48,49]. Tarmann and colleagues [50] reported that brachytherapy with TTT resulted in worse VA results and did not improve tumour control. For small and unclassified lesions with few risk factors, TTT has been reported to be effective as an initial treatment in approximately 80% of cases [51,52]. However, Shields and colleagues [52] advised that, when possible, small choroidal melanomas with multiple risk factors should be managed with treatment modalities other than TTT.

## 14. Choroidal Melanoma Treatment

Management modalities for choroidal melanoma depend on the VA of the involved eye, VA of the other eye, tumour location and size, and existence of extraocular spreading [1]. Factors that were identified to predict choroidal nevus transformation to choroidal melanoma include thickness > 2 mm, SRF, symptoms, orange pigmentation, tumour margin < 3 mm to disk, lack of surrounding, and US hollowness [53]. If the choroidal nevi do not show any features, the patients should be followed up every six months for the first year and then once each year thereafter. If there are one or two features, the patient must be followed up every 4 to 6 months [1].

Small and medium-sized choroidal melanomas may be managed by radiation therapy, laser treatment, and location tumour resection, whereas treatment options for larger choroidal melanomas include enucleation or orbital exenteration, if there is metastasis into the orbit [54].

Patients usually tend to fear death due to cancer and worry about losing vision in the affected eye. They tend to worry about disfigurement that may result after enucleation of the affected eye. Furthermore, they often worry that they may experience loss of vision or blindness in the other eye. Many factors should, therefore, be taken into consideration before determining which treatment approach should be employed as it may affect the patient’s overall quality of life as well as emotional and psychological function [1].

## 15. Management for Metastatic Disease

Even with primary treatment and tight surveillance, approximately 50% of UM patients will have metastases [4]. UM spreads through haematogenous spread, mainly to the liver [7]. Shields and colleagues [55] analysed > 8000 patients and found that 25% of choroidal melanoma patients had developed metastatic disease at 10 years.

The probabilities for secondary spread are related to the size of the initial malignancy at detection time; at 10 years, metastasis was diagnosed in 49% of large melanoma, 26% of medium melanoma, and 12% of small melanoma [55]. The five-year relative survival rate is 70 to 80%, notwithstanding developments in the management of the initial malignancy, with no documentation of reduction in metastatic death [56,57].

Management modalities for metastatic UM are limited. For resectable disease, surgical resection is the main approach [7]. In cases of distant metastatic disease, the main treatment option is systemic chemotherapy [1]. Options include chemo-embolisation, radio-embolisation, and chemotherapeutics agents, which have been shown to have limited efficacy [7]. Studies have reported that there can be 0–15% response rates when using chemotherapeutics agents such as treosulfan, cisplatin, temozolomide, dacarbazine, and fotemustine [2]. A selective kinase inhibitor known as selumetinib was shown to enhance response rates and progression-free survival in metastatic UM patients in contrast to chemotherapy [58]. Nevertheless, there was no improvement in overall survival [58].

Metastatic UM treatment is still a major challenge. Clinical trials are needed for ongoing research including treatments aimed at the PI3K and/or MAPK pathway, and epigenetic alteration with HDAC and DNA methyltransferase inhibitors [59]. Other clinical trials based on guiding the immune system in targeting malignant cells are still under development. IMCgp100 has demonstrated a favourable safety form and tumour shrinkage in metastatic UM [60,61].

## 16. Surveillance

Patients with UM are at lifetime risk of having metastasis even with treatment of primary malignancy [7]. Usual sites of metastases comprise the bones (16%), lungs (24%), and liver (90%) [62,63]. There are no consensus guidelines concerning optimal surveillance tests after initial management. Multiple imaging options such as chest X-ray (CXR), abdominal ultrasound (US), CT, MRI, and PET have been used [4].

Of these, MRI has demonstrated the highest sensitivity in identifying small liver lesions [2]. The National Guidelines for the Management of Uveal Melanoma in the UK advise that non-ionising imaging of the liver (US or MRI) should be conducted for all patients to avoid excessive radiation during metastatic surveillance [64].

Based on gene-expression profiling or cytogenetics, patients with low-risk disease should be considered for routine imaging, including chest CT, and abdomen and pelvis MRI every 6 to 12 months. In contrast, patients with a higher metastatic recurrence risk warrant a tighter follow-up, with imaging taken every 3 to 6 months [2].

## 17. Prognosis and Survival

Nearly 50% of UM patients will suffer from metastases, with the liver being the most common initially affected organ. Around 20–30% of UM patients die within 5 years and 45% die within 15 years [2]. Several prognostic factors are related to metastatic death, such as increasing tumour size, increasing patient age, lymphocytic infiltration, epithelioid cell type, ciliary body involvement, and multiple biomarkers such as human HLA molecules [7,65]. Damato and colleagues [66] stated that the most significant factors that predict metastatic death are chromosome 3 loss, epithelioid cell histopathology, and basal tumour diameter.

Unfortunately, the survival rates of patients suffering from UM have not improved in the past 30 years, even with the availability of various treatment modalities [1]. The percentage of patients with hepatic metastasis is approximately 80%, 92%, and only 1% after one year, two years, and over five years, respectively, regardless of the management option [67].

## 18. Conclusions

Choroidal melanomas are rare, yet it is crucial to be cautious when examining a choroidal mass. Early detection is vital; therefore, fundus examination must be considered as most choroidal melanoma cases are diagnosed clinically. The patient must be educated about the life expectancy, lifetime risk of potential metastases, management modalities, and expected vision outcome.

## Figures and Tables

**Figure 1 medicines-10-00011-f001:**
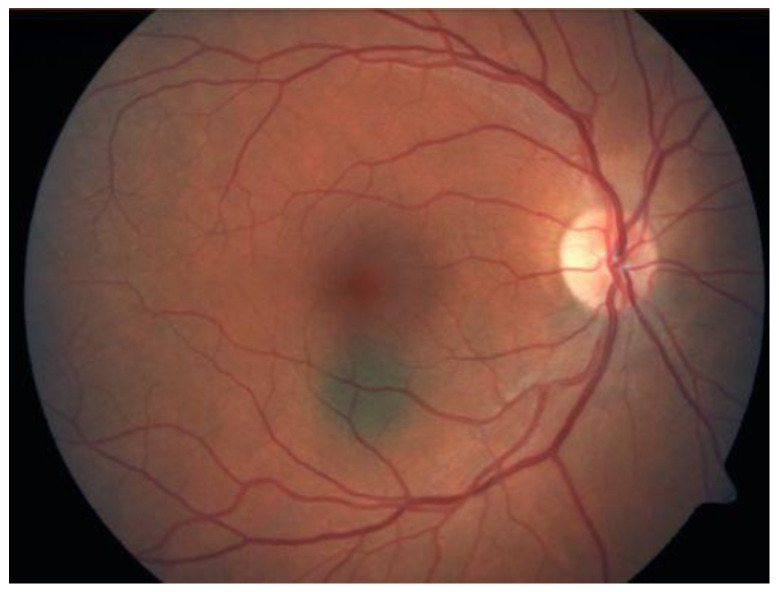
Choroidal nevus.

**Figure 2 medicines-10-00011-f002:**
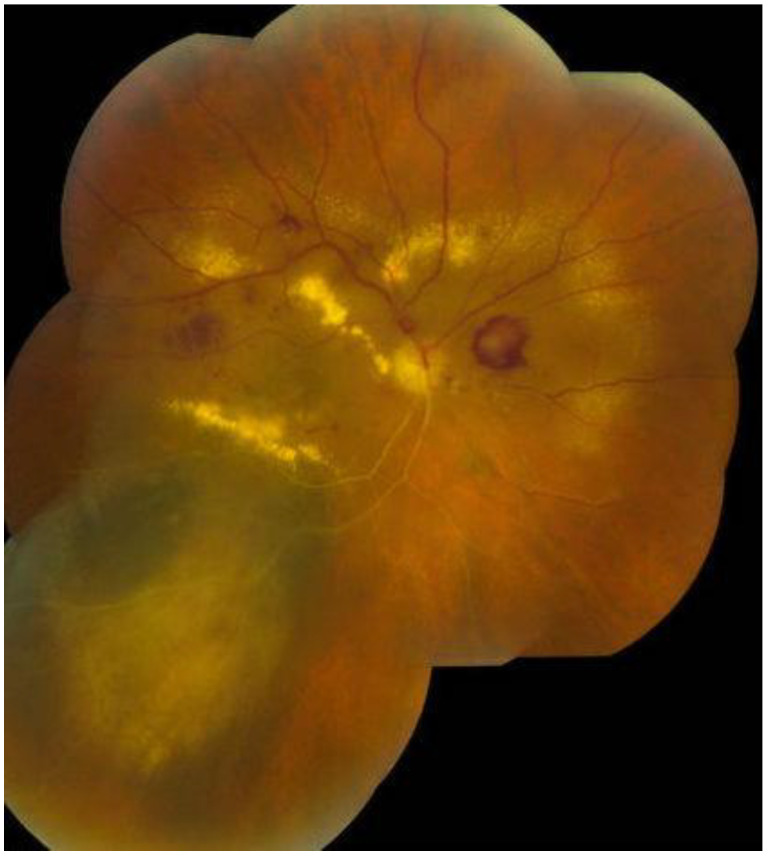
Darkly pigmented melanoma.

**Figure 3 medicines-10-00011-f003:**
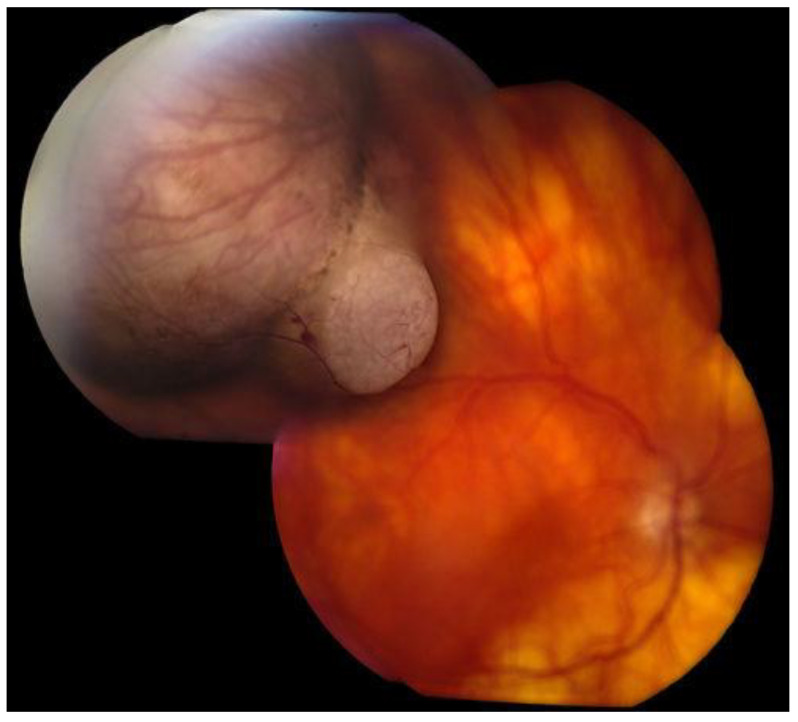
Amelanotic choroidal melanoma.

**Figure 4 medicines-10-00011-f004:**
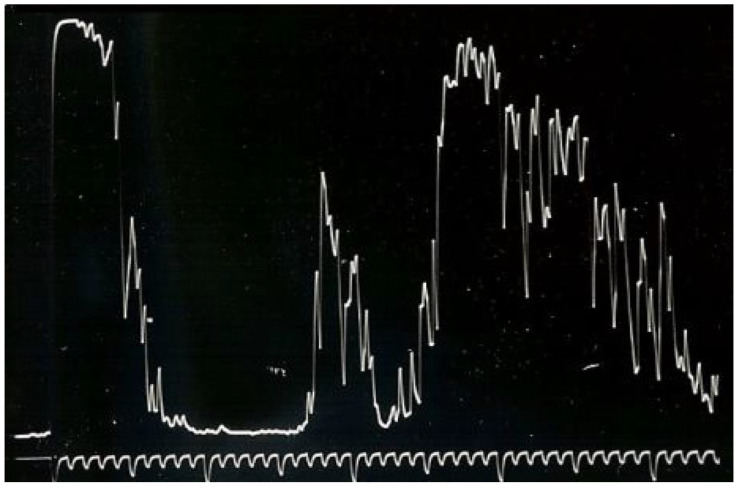
A-scan US reveals medium to low internal reflectivity with vascular pulsations.

**Figure 5 medicines-10-00011-f005:**
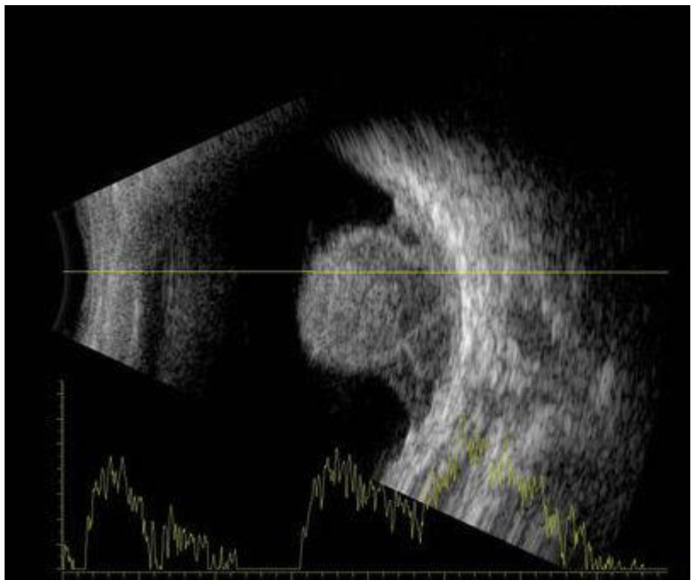
B-scan US shows a mushroom or dome-shaped tumour. The mass is acoustically silent, indicating a dark appearance inside the tumour.

**Figure 6 medicines-10-00011-f006:**
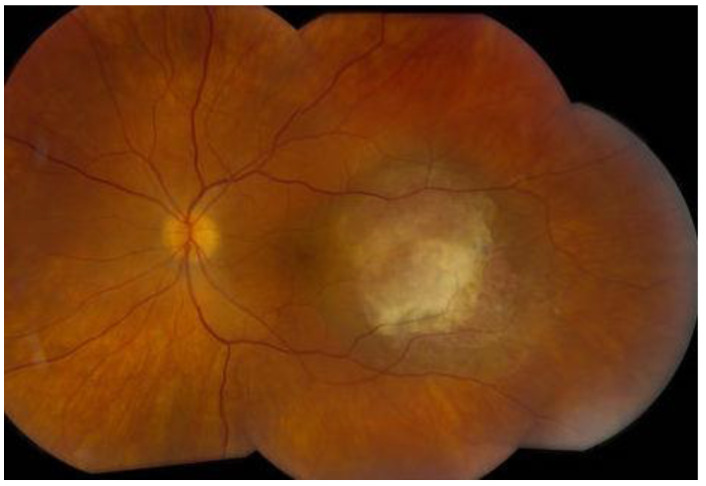
Choroidal haemangiomas.

**Figure 7 medicines-10-00011-f007:**
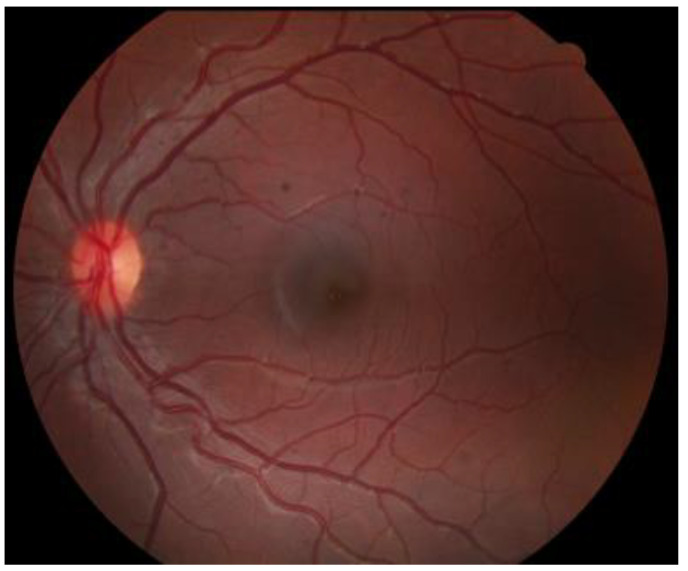
Retinal pigment epithelium hypertrophy.

**Figure 8 medicines-10-00011-f008:**
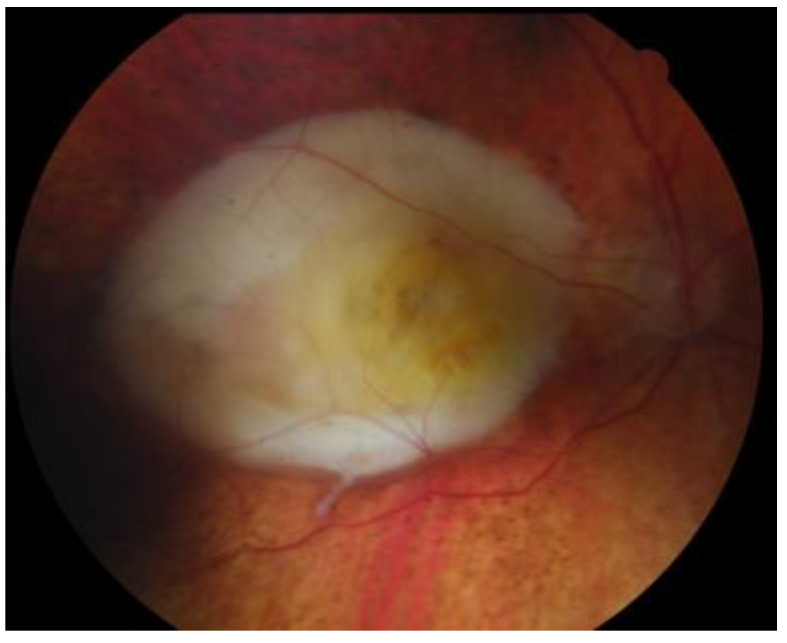
Disciform scar.

**Figure 9 medicines-10-00011-f009:**
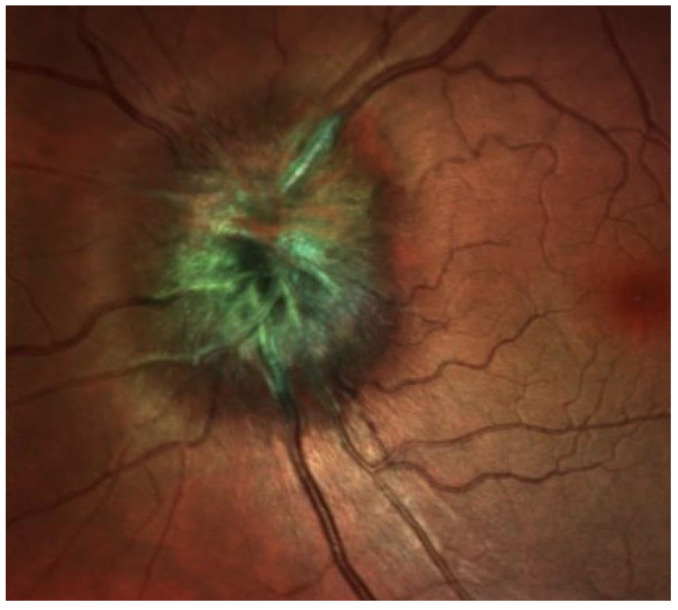
Melanocytoma.

**Figure 10 medicines-10-00011-f010:**
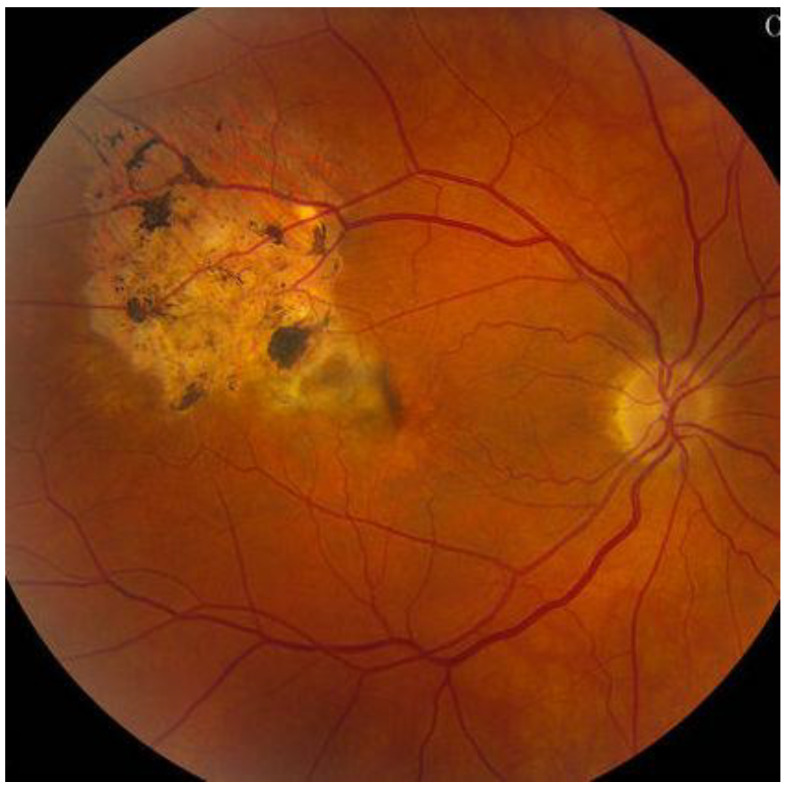
Choroidal osteoma.

**Figure 11 medicines-10-00011-f011:**
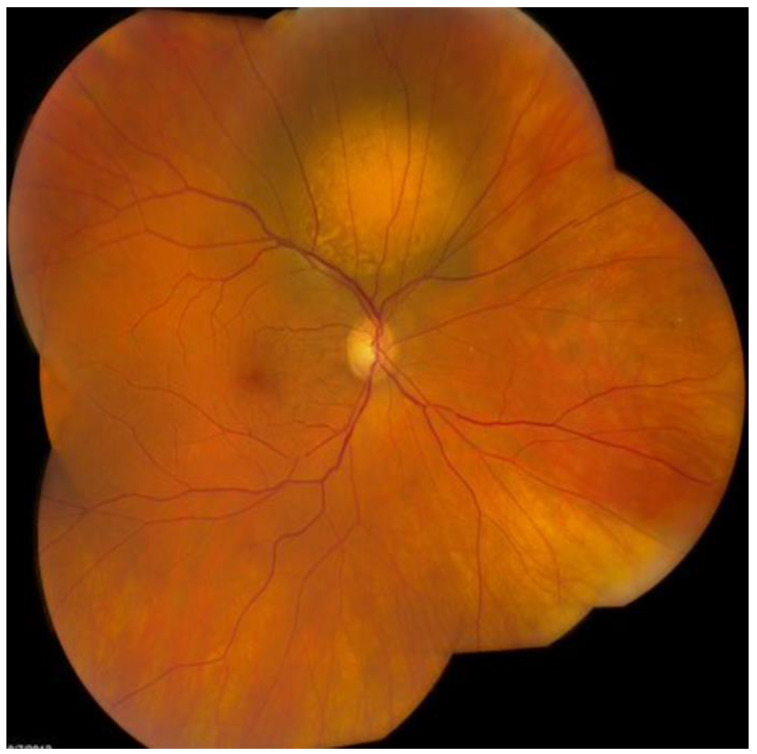
Choroidal metastasis.

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
