# Peer review of "Choroidal Melanoma: A Mini Review"

_medicines, 2023, doi:10.3390/medicines10010011_

Round 1

Reviewer 1 Report

This ocular melanoma mini review provides a comprehensive range of information about choroidal melanoma. However, the mini-review concept inherently means that every concept is not fully discussed, with one or two references provided for some important topics, not covering all the debates present in the literature, or without the possibility to add nuances.

 - " Monosomy of chromosome 3 predicates a mortality rate higher than 50%, while disomy of chromosome 3 has a 100% survival rate [8]" Some others papers do not have this results for patients with disomy of chromosome 3 (ex D3 : g8q ; Cassoux N, Rodrigues MJ, Plancher C, Asselain B, Levy-Gabriel C, Lumbroso-Le Rouic L, Piperno-Neumann S, Dendale R, Sastre X, Desjardins L, Couturier J. Genome-wide profiling is a clinically relevant and affordable prognostic test in posterior uveal melanoma. Br J Ophthalmol. 2014 Jun;98(6):769-74. doi: 10.1136/bjophthalmol-2013-303867. Epub 2013 Oct 29. PMID: 24169649; PMCID: PMC4033183.)

 - "From a clinical point of view, RNA based genetic analysis for choroidal melanoma prognosis, which tests the expression profile for 15 genes seeking to produce prognostic subgroups for metastatic risk, may be done prior to commencing treatment [5]." Ref 5 is a case report about enucleation and therefore can't be a correct reference about this.

- "There is no known cause resulting in UM". In this paragraph, authors can remind that the presence of choroidal naevus or ocular melanocytosis are risk factors for choroidal melanoma, also BAP 1 mutation familial history. Literature should be read again.

 - "Ultraviolet exposure as risk factor". The most recent papers about uveal melanoma contradict this, as only a low burden of mutations is observed in tumoral cells. For exemple : "A role for ultraviolet (UV) radiation exposure inUM has been suggested, as in cutaneous melanoma, but is unlikely to be involved in posterior UMs owing to the very low mutation burden and absence of a UV muta- tional signature in UM5–8; the cornea, lens and vitreous effectively remove most UV radiation such that very little reaches the choroid9 . However, tumour initiation has a predilection for the macula, where light is focused, suggesting a role for non- UV wavelengths10. UMs of the iris are located in front of the lens, and may be under the influence of UV- induced DNA damage (Box 1)" Jager MJ, Shields CL, Cebulla CM, Abdel-Rahman MH, Grossniklaus HE, Stern MH, Carvajal RD, Belfort RN, Jia R, Shields JA, Damato BE. Uveal melanoma. Nat Rev Dis Primers. 2020 Apr 9;6(1):24. doi: 10.1038/s41572-020-0158-0. Erratum in: Nat Rev Dis Primers. 2022 Jan 17;8(1):4. PMID: 32273508.

- "Choroidal melanomas can lead to retinal detachment based on the tumour shape [6]." Also retated to tumoral exsudative detachment.

- Differential diagnosis : choroidal metastases should me mentioned.

- "Transscleral resection has higher recurrence rates in contrast to enucleation or brachy- therapy [4]." Surgical resection is under debates (pre or post irradiation, recurrences, ...). As recurrences (and fatal ones) are the main issue with transscleral resection, correct evaluation of recurrences should be provided.

- "Factors that increase the risk of development and early detection choroidal melanoma include thickness > 2 mm, SRF, symptoms, orange pigmentation, tumour margin < 3 mm to disk, lack of surrounding and US hollowness [29]."  In order to increase the reader's understanding, please specify that these elements quoted concern the risk factors for the transformation of a choroidal nevus into a choroidal melanoma.

- About TTT : I agree that TTT has some indications but the authors should balance the written text with this reference : Mashayekhi A, Shields CL, Rishi P, Atalay HT, Pellegrini M, McLaughlin JP, Patrick KA, Morton SJ, Remmer MH, Parendo A, Schlitt MA, Shields JA. Primary transpupillary thermotherapy for choroidal melanoma in 391 cases: importance of risk factors in tumor control. Ophthalmology. 2015 Mar;122(3):600-9. doi: 10.1016/j.ophtha.2014.09.029. Epub 2014 Nov 13. PMID: 25439431.  

Jerry Shields wrote : "We advise that, when possible, small choroidal melanomas with multiple risk factors be treated with methods other than TTT"

- Metastases risk : different data are provided two times at the beginning of §15 and also at the beginning of § 16 and § 17

 - Ref about IMCgp100 is really not up to date (2016). I incite the authors to adapt their text regarding more recent ref.

- "Unfortunately, survival rates of patients suffering from UM have not altered in the past 30 years, even with the availability of various treatment modalities [1]. Patients with hepatic metastasis have a median survival of 6 months with an approximate survival rates of 10% at 2 years and 15–20% at 1 year, regardless of the management option [42, 43]."

      Ref 42 and 43  : 2005 and 2006 papers ! I incite the authors to rewrite this using more recent refs.

- Reference number 6 is problematic for me. It is frequently quoted in this article. This is a reference from StatPearls, about all ocular melanoma (intra and conjunctival ones), wich very few discussion and poor references.

Authors used this ref n°6 to justify this sentence in their manuscript : " Positron emission tomography (PET), scan is essential if there is a lymphatic involvement [6]."  This is a real mistake.

The authors were not able to distinguish, within this reference n°6, from the elements referring to intraocular tumors, therefore the uveal melanoma mentioned here, and tumors of the conjunctiva, for which a PET-CT is required.

I work in a European reference center and, like all reputable reference centers, we have "never" seen lymphatic dissemination of uveal melanoma and it is very rare to prescribe a PET scan as an extension assessment ((it is however used in case of proven metastatic disease).

Reviewer 2 Report

Soliman et al provide an excellent, succint review of choroidal melanoma. I would suggest two revisions to the paper before its suitable for publication.

One: Cai et al showed the superiority of gene expression profiling vs the Tumor-Node-Metastasis classification. Authors should emphasize that contrary to other tumors; including conjunctival melanoma; choroidal melanoma DOES not behave according to the TNM (seldom metastasizes to the lymph nodes) and include relevant reference points to this. 

Two: A brief description of indeterminate choroidal nevi should also be included, as many ophthalmologists have trouble with their differentiation from true choroidal melanoma, this was shown by Harbour et al as well. This review would be highly suitable as a reference for the general ophthalmologist/retina specialist and this information should be included.

Round 2

Reviewer 1 Report

I thank the authors for having improved the manuscript according to the remarks